# Bactabolize is a tool for high-throughput generation of bacterial strain-specific metabolic models

Ben Vezina[1]*[†], Stephen C Watts[1][†], Jane Hawkey[1], Helena B Cooper[1], Louise M Judd[1], Adam WJ Jenney[2], Jonathan M Monk[3], Kathryn E Holt[1,4], Kelly L Wyres[1]*

[1]Department of Infectious Diseases, Central Clinical School, Monash University, Melbourne, Australia; [2]Microbiology Unit, Alfred Health, Melbourne, Australia; [3]Department of Bioengineering, University of California, San Diego, San Diego, United States; [4]Department of Infection Biology, London School of Hygiene & Tropical Medicine, London, United Kingdom

*For correspondence:
benjamin.vezina@monash.edu (BV);
kelly.wyres@monash.edu (KLW)

[†]These authors contributed equally to this work

Competing interest: The authors declare that no competing interests exist.

**Abstract** Metabolic capacity can vary substantially within a bacterial species, leading to ecological niche separation, as well as differences in virulence and antimicrobial susceptibility. Genome-scale metabolic models are useful tools for studying the metabolic potential of individuals, and with the rapid expansion of genomic sequencing there is a wealth of data that can be leveraged for comparative analysis. However, there exist few tools to construct strain-specific metabolic models at scale. Here, we describe Bactabolize, a reference-based tool which rapidly produces strain-specific metabolic models and growth phenotype predictions. We describe a pan reference model for the priority antimicrobial-resistant pathogen, *Klebsiella pneumoniae*, and a quality control framework for using draft genome assemblies as input for Bactabolize. The Bactabolize-derived model for *K. pneumoniae* reference strain KPPR1 performed comparatively or better than currently available automated approaches CarveMe and gapseq across 507 substrate and 2317 knockout mutant growth predictions. Novel draft genomes passing our systematically defined quality control criteria resulted in models with a high degree of completeness (≥99% genes and reactions captured compared to models derived from matched complete genomes) and high accuracy (mean 0.97, n=10). We anticipate the tools and framework described herein will facilitate large-scale metabolic modelling analyses that broaden our understanding of diversity within bacterial species and inform novel control strategies for priority pathogens.

## eLife assessment

This study presents Bactabolize, a **valuable** tool for the rapid genome-scale reconstruction of bacteria and the prediction of growth phenotypes. Using validated methodology, the tool relies on a reference pan-genome model to create strain-specific draft metabolic models, as demonstrated in this study using *Klebsiella pneumoniae*. While the evidence in this specific case is **solid**, validation across diverse bacterial species is yet to be confirmed.

## Introduction

Bacteria exhibit metabolic diversity and can utilise a broad range of substrates for growth. It has become clear amongst pathogens that there is an intertwined relationship between metabolism and nutrient usage with virulence and antimicrobial resistance (*Su et al., 2016*; *Wu et al., 2021*; *Mir et al.,*

*2014*; *Vornhagen et al., 2019*; *Eberl et al., 2021*; *Jenior et al., 2021*; *Hudson et al., 2022*). Comparative analyses of metabolic profiles (e.g. substrate usage) are key to fully understanding these relationships. Traditionally, these profiles have been assessed via phenotypic growth on a limited number of substrates, such as those used to delineate between species (*Rodrigues et al., 2019*; *Blin et al., 2017*; *Brisse et al., 2009*) which form the basis of a number of commercial products for species identification. However, these methods are not sufficiently discriminatory for in-depth comparisons within species, and alternative approaches such as the Omnilog Phenotype MicroArray system (Biolog) are too expensive and/or labour intensive for application to large numbers of isolates. Similarly, probing of essential metabolism-associated genes via transposon mutant libraries (e.g. to identify novel virulence factors and therapeutic targets) (*Vornhagen et al., 2019*; *Mobegi et al., 2014*; *Hogan et al., 2018*) cannot be easily scaled across diverse bacterial populations.

Genome-scale metabolic models or metabolic reconstructions are a computational approach to analysing the metabolic potential of an organism, within which the entire biochemical network is represented as a stoichiometric matrix (*Edwards and Palsson, 1999*). Metabolic models are constructed programmatically, but typically informed and at least partially validated using phenotypic growth data (*Hawkey et al., 2022*; *Henry et al., 2017*; *Liao et al., 2011*). Once constructed, they can be run through simulations and analysed under various contexts, such as in silico growth experiments (flux balance analysis [FBA]) to predict substrate usage profiles (*Orth et al., 2010*), evaluate the impact of single-gene knockouts on growth (*Hawkey et al., 2022*; *Stanway et al., 2019*), and identify metabolic chokepoints for drug targets (*Ramos et al., 2018*), among others. Traditionally, metabolic models are strain-specific (i.e. each model represents a unique individual http://bigg.ucsd.edu/models) and may not be applicable to other isolates due to unrepresented genetic diversity.

We recently described 37 curated strain-specific models for the *Klebsiella pneumoniae* species complex (*Kp*SC) (*Hawkey et al., 2022*) comprised of *K. pneumoniae* and its close relatives (*Wyres et al., 2020*). These organisms are a common cause of healthcare-associated infections worldwide, and among the World Health Organization's priority antimicrobial resistant pathogens (*WHO, 2017*). *Kp*SC are highly diverse and gene content can differ substantially between strains (*Gorrie et al., 2022*; *Holt et al., 2015*). Accordingly, our models varied in terms of gene and reaction content, resulting in variable growth substrate usage profiles and metabolic redundancy (*Hawkey et al., 2022*). Similar variation has also been described in other key bacterial pathogens, for example *Escherichia coli* (*Monk et al., 2013*), *Salmonella enterica* (*Seif et al., 2018*), *Staphylococcus aureus* (*Bosi et al., 2016*), and *Pseudomonas aeruginosa* (*Bartell et al., 2017*). This is highly relevant to the use of metabolic models for the exploration of virulence and antimicrobial resistance, and for the identification of novel drug targets. Therefore, such works should seek to include multiple strain-specific models, and in some cases 100s–1000s of models may be required to accurately represent population diversity (*Gorrie et al., 2022*; *Croucher et al., 2014*; *Cummins et al., 2022*).

There are several open source tools currently available that can rapidly produce strain-specific metabolic models, including CarveMe (*Machado et al., 2018*), gapseq (*Zimmermann et al., 2021a*), ModelSEED (*Seaver et al., 2021*), and KBase (*Arkin et al., 2018*) (see the recent review by Mendoza and colleagues for comparative descriptions *Mendoza et al., 2019*), as well as a recently published modelling and analysis pipeline, ChiMera, which leverages CarveMe for model construction (*Tamasco et al., 2022*). In their systematic analysis, Mendoza et al. indicated CarveMe and ModelSEED to be of particular interest for large-scale studies due to their speed and model quality (*Mendoza et al., 2019*) (note that gapseq was published later and therefore not included in this review). KBase implements ModelSEED (*Seaver et al., 2021*) as a web interface application, limiting its utility for high-throughput analysis of 100s–1000s of bacterial genomes. CarveMe is a command line application; it is open source but is dependent on commercial solvers such as CPLEX (free for academic use). However, its use of a universal reference model may limit specificity of strain-specific models (*Norsigian et al., 2020*), and result in overestimation of model genes. These limitations can be overcome by manual curation of the output models, but such curation is highly labour intensive and not suitable for high-throughput analyses. Furthermore, the CarveMe database (BiGG universal_model) appears to be no longer actively maintained, meaning that there is no opportunity to integrate novel structural and/or biochemical data as these become available in the literature (as discussed in COBRA community forums). gapseq is a recently published command line tool which leverages an independent universal database (*Zimmermann et al., 2021a*). In their comparative analyses, the gapseq authors demonstrate

superior accuracy to both CarveME and ModelSEED; however the concerns about the specificity of universal models remain and it has been reported that model construction takes considerable time (several hours in some cases *Zimmermann et al., 2021b*).

Here, we present Bactabolize (*Watts et al., 2023*), an easy-to-use tool which allows scalable production of strain-specific draft metabolic models and prediction of growth phenotypes in under 3 min per input genome. Bactabolize builds upon the reference-based model reconstruction approach described by *Norsigian et al., 2020*, leveraging the COBRApy framework (*Ebrahim et al., 2013*) and BiGG nomenclature (*Schellenberger et al., 2010*). We present a pan-metabolic reference model for the *Kp*SC (derived from our 37 curated strain-specific models *Hawkey et al., 2022*), and describe an exemplar quality control framework for the application of Bactabolize to *Kp*SC draft genome assemblies. We show that Bactabolize can rapidly produce strain-specific models from draft genomes with a high degree of completeness (as compared to models generated from completed genome assemblies), resulting in highly accurate growth predictions that match or exceed the accuracy of models from CarveMe and manual curation efforts.

## Results
### Description of Bactabolize

Bactabolize is written in Python 3 and utilises the metabolic modelling library COBRApy (*Ebrahim et al., 2013*). Bactabolize has four main commands:

1. Draft model generation (*draft_model* command), which generates a strain-specific draft metabolic reconstruction ('model') using the approach outlined previously (*Norsigian et al., 2020*), and uses gap-filling to identify any missing reactions required to simulate growth in the user-specified conditions
2. Patching incomplete models (*patch_model* command) by the addition of missing reactions, for example those identified by the automated gap-filling process
3. Substrate usage analysis via FBA (*fba* command) to predict growth outcomes for a specified range of substrates supported by the model(s) *Figure 1*.

Additional processing scripts are provided alongside Bactabolize to improve model metadata annotation (improve_model_annotations.py), convert models generated using KBase and Model-SEED to Bactabolize/BiGG-compatible format (SEED_to_BiGG_model_convert.sh), generate network graph files from models (model_to_network_graph.py) and merging output FBA profiles (merge_fba_profiles_longtable.sh). Full documentation including example code and test data are available at the Bactabolize code repository (*Watts et al., 2023*).

For draft model construction, Bactabolize requires users to provide an input assembly (annotated or unannotated FASTA or Genbank format respectively), a reference model (JSON format), and the corresponding reference sequence data (gene and protein sequences in two separate multi-fasta files or a single Genbank annotation in a .gbk file) (*Figure 1—figure supplement 1*). For optimum results we suggest using a pan-model that captures as much diversity as possible for the target species or group of interest, because Bactabolize's reconstruction method is reductive, that is each output strain-specific model will include only genes, reactions, and metabolites that are present in the reference or a subset thereof (although novel genes, reactions, and metabolites can be added via manual curation).

If the input assembly is unannotated, Bactabolize will identify coding sequences using Prodigal (*Hyatt et al., 2010*) but will otherwise honour the existing coding sequence (CDS) notations and optionally use Prodigal to search for additional CDS. Draft genome-scale metabolic models are output in both SMBL v3.1 (*Keating et al., 2020*) and JSON formats (one pair of files for each independent strain-specific model), along with an optional MEMOTE quality report (*Lieven et al., 2020*). Bactabolize will identify orthologs in the input genome(s) compared to the reference sequence data using Bi-directional BLAST (*Camacho et al., 2009*) Best Hits (BBH) (*Hernández-Salmerón and Moreno-Hagelsieb, 2020*) using BLAST+ (*Norsigian et al., 2020*). Users can parameterise the ortholog finding settings (coverage and identity thresholds) for BBH. Alternatively, there is the option of using protein similarity to identify orthologs instead of identity.

Once a draft model has been constructed, it is validated via a simulated growth experiment on user-input choice of media and atmosphere (aerobic or anaerobic). Predefined media include BG11 (Gibco), M9+glucose (*Norsigian et al., 2020*), nutrient media (*Zimbro et al., 2009*), Luria-Bertani (LB) (*Zimbro*

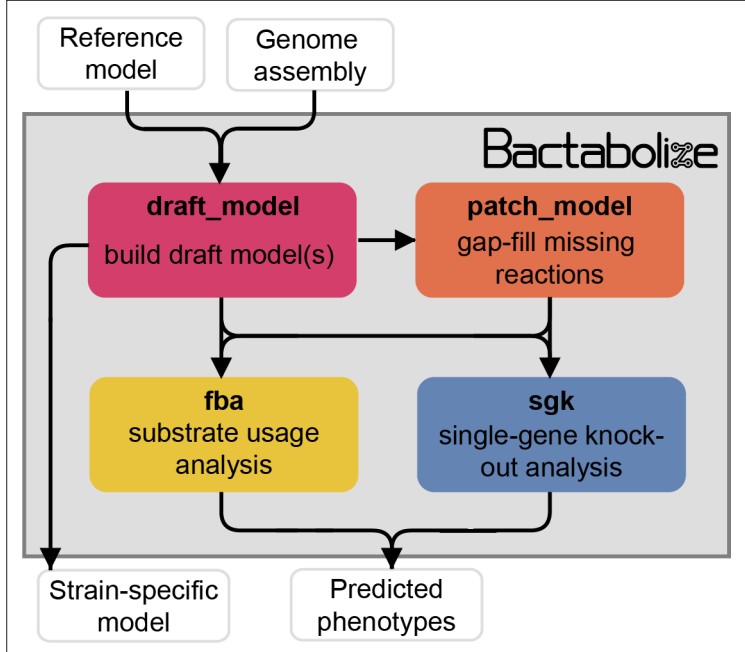

**Figure 1.** Simplified overview of Bactabolize's main commands. In pink is the *draft_model* command, which builds a draft strain-specific metabolic model using an input reference model and an input target assembly (approach adapted from **Norsigian et al., 2020**). If the model fails to simulate growth, Bactabolize will attempt automated gap-filling and produce a model patch file. The *patch_model* command (orange) allows the addition of missing reactions to produce a valid draft model that can simulate growth in a user-specified growth environment. A functioning model can be passed to the *fba* command (yellow), which performs Flux Balance Analysis to simulate growth in the user specified conditions, across all carbon, nitrogen, phosphorus, and sulphur metabolite sources supported by the model under aerobic and anaerobic conditions. The *sgk* command (blue) shows the single-gene knockout analysis, which outputs a predicted phenotype. User inputs and outputs are shown in white boxes while Bactabolize commands are shown inside the grey box. Additional graphics can be found in **Figure 1—figure supplements 1–4**.

The online version of this article includes the following figure supplement(s) for figure 1:

**Figure supplement 1.** Flow diagram showing the overview of the draft_model module from Bactabolize, which produces draft metabolic models.

**Figure supplement 2.** Flow diagram showing the overview of the patch_model module from Bactabolize, which patches metabolic models that do not simulate growth.

**Figure supplement 3.** Flow diagram showing the overview of the fba module from Bactabolize, which performs growth simulations using Flux Balance Analysis.

**Figure supplement 4.** Flow diagram showing the overview of the sgk module from Bactabolize, which performs single-gene knockout analysis.

---

*et al., 2009*), Tryptic Soy (TSA) (*Zimbro et al., 2009*), TSA+sheep blood (*Zimbro et al., 2009*), LB as specified by the CarveMe developers (*Machado et al., 2018*), Chemically Defined Medium-like (*Mendoza et al., 2019*), Plantarum Minimal Medium (PMM) PMM5-like (*Mendoza et al., 2019*) and PMM7-like (*Mendoza et al., 2019*). Users can also define custom media as Bactabolize supports several complex media ingredients, including peptone (peptic digest of bovine and porcine tissue) (*BD Biosciences, 2015*; *Loginova et al., 1974*; *ThermoFisherScientific, 2019*), tryptone (pancreatic digest of casein) (*BD Biosciences, 2015*; *ThermoFisherScientific, 2019*; *Clausen et al., 1985*), soy peptone/soytone (digest of soymeal) (*BD Biosciences, 2015*; *ThermoFisherScientific, 2019*; *Hagely et al., 2013*; *Choct et al., 2010*), yeast extract (*Tomé, 2021*; *Plata et al., 2013*; *Liu et al., 2018*; *Blagović, 2001*; *Blagović et al., 2005*; *Avramia and Amariei, 2021*), and beef extract (*BD Biosciences, 2015*; *ThermoFisherScientific, 2019*). If the model fails to simulate growth, gap-filling is performed to indicate missing reactions. Users can add these reactions to a patch JSON file and optionally use the *patch_model* command to correct the model (*Figure 1—figure supplement 2*).

Bactabolize uses a conservative gap-filling approach that only adds the minimum number of reactions to enable growth under the chosen conditions. We recommend testing the models in minimal media and atmosphere expected to support growth for all isolates of the species of interest, unless the user has access to matched phenotypic data demonstrating growth for individual isolates in specific conditions. Aggressive gap-filling will effectively homogenise the models and should be avoided if the goal is to understand the underlying strain diversity.

Substrate usage analysis (the *fba* command) is performed iteratively for each possible carbon, nitrogen, sulphur, and phosphor substrate supported by the model(s) (*Figure 1—figure supplement 3*), by replacing the default substrate in the user-specified growth medium (specified in the fba_spec JSON file). For example, in M9 media the default substrates are glucose (carbon), ammonia (nitrogen), sulphate (sulphur), and phosphate (phosphor). Each substrate can be tested in aerobic and/or anaerobic conditions. Growth prediction output is recorded in a tab delimited file (one per strain). The merge_fba_profiles_longtable.sh helper script will combine the outputs for multiple strains into a single file for downstream analysis.

The growth impacts of single-gene knockout mutations can be simulated via the *sgk* command (*Figure 1—figure supplement 4*). Bactabolize will iterate through every gene in the model, temporarily removing it and its associated reactions (unless they are also associated with another gene) and running FBA to simulate growth in the user-specified conditions. The output is comparable to single-gene knockout studies such as transposon mutagenesis and can be used to probe gene essentiality.

## *Kp*SC pan-metabolic reference model

We constructed a species complex-specific pan-metabolic reference model by combining a collection of 37 manually curated models for which we have previously demonstrated high accuracy (range 88.3–96.8% for prediction of 94 distinct growth phenotypes, *Hawkey et al., 2022*). These models represent a diverse collection of *Kp*SC (*Hawkey et al., 2022*) (including at least one each of the seven major taxa in the complex; *K. pneumoniae, Klebsiella variicola* subsp *variicola, Klebsiella variicola* subsp *tropica, Klebsiella quasipneumoniae* subsp *quasipneumoniae, Klebsiella quasipneumoniae* subsp *similipneumoniae, Klebsiella quaisivariicola, Klebsiella africana*). The combined pan-model, known as *Kp*SC-pan v1, comprises a total of 1265 distinct genes, 2319 reactions, and 1696 metabolites, and is available on GitHub (*Vezina et al., 2023*).

## Performance comparison

We compared the output and performance of Bactabolize to the two previously published tools that can support high-throughput analyses, that is CarveMe (*Machado et al., 2018*) and gapseq (*Zimmermann et al., 2021a*). To aid interpretation in the context of community standard approaches, we also include a comparison to the popular web-based reconstruction tool, KBase (ModelSEED), and a manually curated metabolic reconstruction of *K. pneumoniae* strain KPPR1 (also known as VK055 and ATCC 43816, metabolic model named iKp1289) (*Henry et al., 2017*). This isolate was chosen as there is a completed genome sequence (Genbank accession: CP009208), single-source growth phenotype (*Henry et al., 2017*), and single-gene knockout growth essentiality data available (*Short et al., 2020*). De novo draft models for strain KPPR1 were built using: (i) Bactabolize with the *Kp*SC pan v1 reference; (ii) CarveMe, with its universal reference model (CarveMe universal); (iii) CarveMe, with *Kp*SC-pan v1 reference (CarveMe *Kp*SC pan); (iv) gapseq; and (v) KBase (ModelSEED). Importantly, neither *K. pneumoniae* KPPR1 nor its genetic lineage (seven gene multi-locus sequence type, ST493) are represented in the *Kp*SC pan reference model, meaning these benchmarking comparisons were on equal footing. Subsequently, each model was used to predict growth phenotypes: (i) in M9 minimal media with different sole sources of carbon, nitrogen, phosphorus, and sulphur; and (ii) for all possible single-gene knockouts in LB under aerobic conditions. The predicted phenotypes were compared directly to the published phenotype data.

Among the high-throughput approaches, the Bactabolize draft model captured fewer genes and reactions (n=1233 and 2307, respectively) than the gapseq (n=1489 and 3186, respectively) and CarveMe universal models (n=1960 and 2857, *Figure 2A*). The Bactabolize and CarveMe universal models contained similar numbers of metabolites (1696 vs 1737, respectively), while the gapseq model contained many more (n=2519). Notably, the initial gapseq model was gap-filled by simulation of growth in M9 minimal media with glucose (consistent with Bactabolize and CarveMe media recipes),

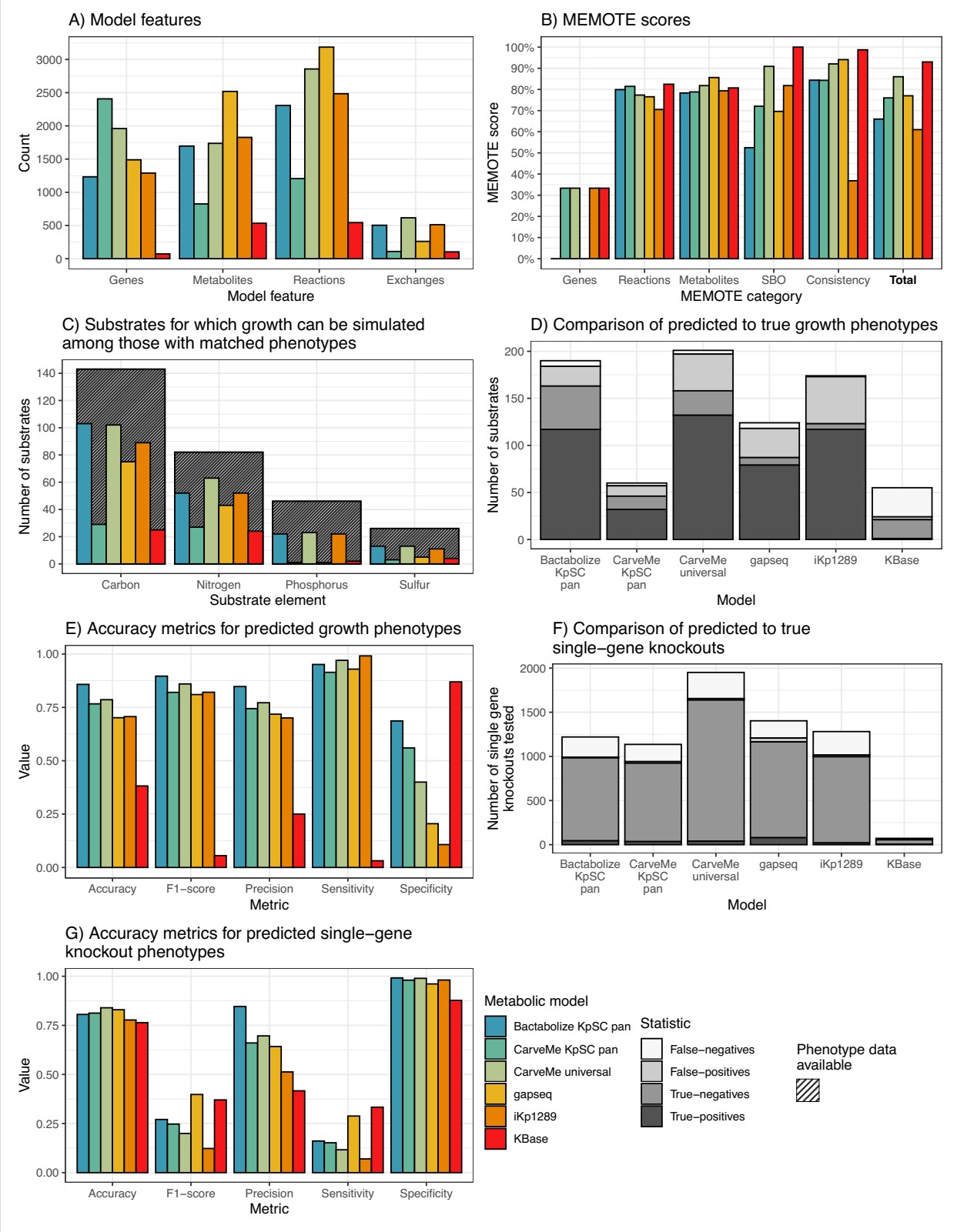

**Figure 2.** *K. pneumoniae* KPPR1 metabolic model benchmarking comparisons. (**A**) Counts of model features: genes, metabolites, and reactions captured by each model. Exchanges refer to number of exchange reactions, a subset of reactions involved in substrate uptake, which determine the number of distinct growth substrates for which phenotypes can be predicted with the model. (**B**) MEMOTE scores indicating the richness of annotations and metadata for metabolic model features according to database outlinks. SBO refers to score of Systems Biology Ontology (SBO), a controlled

*Figure 2 continued on next page*

*Figure 2 continued*

vocabulary for systems biology. Consistency refers to the score of stoichiometric consistency and chemical formulae annotation. Total refers to total MEMOTE score, as a combination of all previous scores, and is shown in bold. (**C**) Counts of carbon, nitrogen, phosphorus, and sulphur growth substrates that can be simulated by models and for which matched phenotypes were available for comparison (*Henry et al., 2017*). Hatched columns indicate the total number of substrates for which phenotypic data for *K. pneumoniae* KPPR1 were described (*Henry et al., 2017*). (**D** and **E**) Accuracy metrics for predicted to true phenotypes for the growth substrates shown in D and E, respectively. False-negatives, true-negatives, false-positives, and true-positives are coloured as shown in legend. (**F** and **G**) Accuracy metrics for the KPPR1 single-gene knockout mutant library described in *Short et al., 2020* as shown in F and G, respectively. Numbers of true-positives and false-positives are shown to the left of the respective columns. *Figure 2—source data 1* and *Figure 2—figure supplements 1–2* contain additional data.

The online version of this article includes the following source data and figure supplement(s) for figure 2:

**Source data 1.** Data table showing single-gene knockout summary of each model.

**Figure supplement 1.** Venn diagram comparing the metabolite output of the best-performing tools for *K. pneumoniae* KPPR1.

**Figure supplement 2.** Venn diagram comparing the reactions output of the best-performing tools for *K. pneumoniae* KPPR1.

which resulted in the addition of 31 reactions to the draft model. This is in contrast to CarveMe and Bactabolize-generated models which produced biomass in M9 minimal media plus glucose without additional gap-filling. The CarveMe *Kp*SC pan model captured considerably more genes than any of the other models (n=2407), but these were associated with many fewer unique reactions and metabolites (1206 and 825, respectively). Upon further investigation we determined that this method resulted in the over-prescription of gene reaction rules (GPRs) to multiple reactions (mean 2.2 GPRs per reaction when compared to Bactabolize using the same pan reference model: 1.94 GPRs per reaction).

*Figure 2—figure supplements 1 and 2* show the overlaps of metabolites and reactions between the high-throughput reconstruction methods after processing with MetaNetX (*Moretti et al., 2021*) to standardise the reaction and metabolite nomenclatures (excluding CarveMe pan for simplicity and given the likely problems of reaction oversubscription). The majority of the reactions included in the Bactabolize model were conserved in either the CarveMe universal model (n=1225, 53.2%), gapseq model (n=54, 2.3%), or both (n=665, 28.9%). The reaction overlap was skewed to the CarveMe universal model which shared 1225 reactions that were conserved in the Bactabolize model but absent from the gapseq model. Notably, the gapseq model contained a large number (2200) of unique reactions (70.4% of those in the model). Similarly, the vast majority of metabolites in the Bactabolize model were conserved in one or both of the other models (n=917, 85.6%). However, it is likely that true overlaps between methods are underrepresented due to the different reaction identifiers and chemical synonyms used within the BiGG (Bactabolize, CarveMe) vs ModelSEED nomenclatures (gapseq), which are difficult to harmonise in an automated manner even after the application of MetaNetX.

In comparison to the low-throughput reconstruction approaches, the Bactabolize model contained a similar number of genes, reactions, and metabolites to the manually curated model (n=1289, 2484, and 1827, respectively) and many more than the KBase model (72, 544, and 534, respectively). This is likely due to the low number of metabolism-associated genes identified by KBase, which has impacted the associated reaction and metabolite data.

MEMOTE scores (produced by the MEMOTE report, *Lieven et al., 2020*) indicate the quality of the model metadata annotations, with the scores ranging between 0% and 100%. These provide a measure of model portability and the level of connected databases available to support the metabolite, reaction, and genetic information represented in the model, but bear no reflection on model accuracy. Bactabolize performs on the lower end, with CarveMe universal and gapseq performing the best (*Figure 2B*). The KBase model appears to perform well in this regard, however this is due to the low number of genes, reactions, and metabolites included in the model. Bactabolize using the *Kp*SC-pan model outperforms the model propagation mode of CarveMe using the same reference model (*Figure 2B*). Work is ongoing to improve the metadata annotations in the *Kp*SC-pan reference model, to support large-scale model propagation.

We assessed the performance of each model for in silico prediction of growth phenotypes compared to the previously published experimental data (*Henry et al., 2017*). Accuracy, sensitivity, specificity, precision, and F1 scores were calculated (*Powers, 2020*). Note that the specific set of growth substrates and gene knockouts that can be simulated is determined by the sets of genes and metabolites captured by each model and is therefore model-dependent (*Source data 1* and *Figure 2—source data 1*). Among those with matched experimental phenotype data, the Bactabolize and

CarveMe universal models were able to predict growth for a greater number of carbon, nitrogen, phosphorous, and sulphur substrates than gapseq, CarveMe *Kp*SC pan, KBase, and iKp1289 models (*Figure 2C*, *Source data 1*). While the CarveMe universal model had the highest number of true-positive growth predictions overall (n=132 of 617 total predictions), it also had a comparably high number of false-positive predictions (n=39 of 617 total predictions, *Figure 2D*). Similarly, the gapseq and iKp1289 models resulted in 31 (262 total predictions) and 50 (513 total predictions) false-positive predictions, respectively. In contrast, the Bactabolize model had fewer false-positive predictions (n=21 of 505 total predictions) alongside a high number of true-positive predictions (n=117 of 505 total predictions), resulting in the highest overall accuracy metrics (*Figure 2E*, *Source data 1*). The KBase model was a notable outlier, associated with a high number of false-negative predictions (n=31 of 103 total predictions) and low false-positive predictions (n=3 of 103 total predictions), presumably resulting from the very low number of genes and reactions included in the model, driving low sensitivity and accuracy.

The gene essentiality results showed that gapseq produced the highest absolute number of true-positive gene essentiality predictions (n=79 of 1403), followed by Bactabolize *Kp*SC pan (n=44 of 1220 total predictions), then CarveMe universal (n=39 of 1951 total predictions). CarveMe universal had the largest number of true-negatives by a wide margin (n=1599 of 1951 total predictions), followed by gapseq (n=1085 of 1403 total predictions), then Bactabolize *Kp*SC pan (n=939 of 1220 total predictions), driving their high accuracies (83.96%, 82.96%, and 80.57%, respectively). The Bactabolize model was associated with the greatest overall precision and specificity (*Figure 2F and G*) while the gapseq model resulted in the highest F1 score and sensitivity.

While model features and accuracy are essential metrics for comparison, computation time is also a key consideration for high-throughput analyses. We recorded the time required for each tool to build draft models for 10 of the completed *Kp*SC genomes used in the quality control framework (see below) on a high-performance computing cluster (Intel Xeon Gold 6150 CPU @ 2.70 GHz and 155 GB of requested memory on a CentOS Linux release 7.9.2009 environment). CarveMe *Kp*SC pan was the fastest with a mean of 20.04 (range 19.90–20.18) s, followed by CarveMe universal at 30.28 (range 29.20–31.80) s, then Bactabolize *Kp*SC pan at 98.05 (range 92.19–100.4) s. KBase took 183.50 (range 120.00–338.00) s per genome via batch analysis, including genome upload time and queuing. gapseq took 5.46 (range 4.55–6.28) hr to produce draft models (not including the required gap-filling), consistent with previous reports (*Zimmermann et al., 2021b*).

## Quality control framework for input genome assemblies

There are now thousands of bacterial genomes available in public databases, the majority of which are in draft form, comprising 10s–1000s of assembly contigs. This fragmentation of the genome is caused by repetitive sequences that cannot be resolved by the assembly algorithm and/or sequence drop-out. The latter can result in the loss of genetic information such that some portion of genes present in the underlying genome are lost from the genome assembly (either completely or partially). This in turn poses a limitation for the reconstruction of metabolic models using these assemblies, since most published approaches use sequence searches to predict the presence/absence of genes and their associated enzymatic reactions. Therefore, if we are to use public genome data for high-throughput metabolic modelling studies, it is essential to evaluate the expected model accuracies and understand the minimum input genome quality requirements.

Here, we performed a systematic analysis leveraging our published curated *Kp*SC models (n=37, *Hawkey et al., 2022*), which were generated using completed genome sequences and were therefore considered to represent 'complete' models for which the underlying genome sequence contains all genes that are truly present in the genome (note the biological accuracy of these models was reported previously [*Hawkey et al., 2022*] and is not the subject of the current study). We randomly subsampled the corresponding Illumina read sets to various depths (10–100×, increments of 10) in triplicate and generated draft assemblies that were passed to Bactabolize for generation of draft metabolic models (*Figure 3—source data 1*). Due to low read depth (≤30×), two isolates were removed from this analysis. Additionally, ten 10× depth read samples failed to produce assemblies, leaving 1040 draft genomes for analysis. The resulting draft metabolic models were compared to the complete models to: (i) determine the proportions of complete model genes and reactions captured in the draft models; and (ii) compare 846 in silico aerobic growth predictions in M9 minimal media,

where growth on 266 carbon, 153 nitrogen, 59 phosphorus, and 25 sulphur sources were examined. Substrates containing multiple elements were tested as sole sources of each element independently and in combination, for example 1,5-diaminopentane was tested as a sole carbon, sole nitrogen, and sole carbon plus nitrogen source.

As expected, assembly quality generally increased with increasing sequencing depth, that is assemblies generated from higher depth read sets were associated with higher N50 values, fewer contigs, and fewer assembly graph dead-ends, although the rate of improvement drastically declined beyond 40–50× depth (*Figure 3—figure supplement 1*, *Figure 3—source data 1*). We noted that it was rare for draft models to capture 100% of model genes and reactions (just 420 of all 1040 draft assemblies were associated with models that captured 100% model genes) (*Figure 3—source data 1*, *Figure 4—figure supplement 1*), even when using the highest quality draft genomes. However, ≥99% of genes and reactions were commonly captured, which plateaued from 40× depth onwards (*Figure 4—figure supplement 1*). Therefore, we sought to evaluate whether ≥99% model capture would produce functionally accurate models.

We used FBA to simulate substrate growth profiles for the 40× depth assemblies, representing a sequencing depth that can be routinely achieved with standard Illumina library preparations. All but one assembly triplicate set (isolate SB4767 98% gene capture, 99% reaction capture) captured ≥99% but ≤100% model genes and/or reactions. The substrate growth profiles were then compared to those of the complete models. The vast majority of draft models produced accurate growth predictions; 102 of 108 models resulted in predictions with 100% concordance to those from the corresponding complete models. Three models for *K. quasipneumoniae similipneumoniae* isolate SB164 resulted in predictions with a mean of 99.8% concordance. The remaining three models were for isolate SB4767 and resulted in mean of 80.4% concordance. Notably, these models were those representing <99% gene capture. Together, these data suggest that draft models capturing ≥99% of the complete model genes/reactions generate highly accurate growth predictions and that these capture rates can be readily achieved from draft genome assemblies.

We investigated the relationships between assembly quality metrics and model gene/reaction capture in more detail. Variation in assembly graph dead-ends accounted for the greatest amount of variation in model capture, closely followed by raw contig counts (cubic polynomial fit, $R^2$ of ≥0.98 for graph dead-ends, $R^2$ of ≥0.9 for contig count). A segmented linear model was fitted to N50 length ($R^2$ ≥0.83), producing a breakpoint at 25,153 bp (*Figure 3*).

To further explore the optimum thresholds for assembly metrics, we tallied the number of draft assemblies resulting in ≥99% and<99% gene and reaction capture at increasing graph dead-end and contig count cut-offs, and decreasing N50 cut-offs. Draft models that captured ≥99% of the complete model genes/reactions were considered 'good' models, whereas draft models that captured <99% of complete model genes/reactions were considered 'bad' models. The optimum threshold for assembly graph dead-end was determined to be ≤200. At this value, 94.44% of 'good' models were captured, and 0% 'bad' models. The optimum threshold for contig counts was determined as ≤130 contigs at which 67.92% of 'good' and 0% 'bad' models were captured (*Figure 4*). The optimum threshold for N50 was determined to be ≥65,000, at which 94.97% of 'good' and 1.71% of 'bad' models were captured. The assembly graph dead-end threshold results in comparatively higher sensitivity (i.e. a higher proportion of 'good' models pass the threshold) than contig count and comparatively better specificity (i.e. lower proportion of 'bad' models pass the threshold) than N50, but the underlying metric information is not universally available because many isolate genomes are deposited in public databases only as assemblies without the associated assembly graph. We therefore recommend a three-tier approach, whereby the assembly graph dead-end criterion is preferred if available, followed by N50 and then contig count.

## Impact of gap-filling models

Of the 901 draft genome assemblies which passed our QC criteria (≤200 assembly graph dead-ends), 23 of the resulting draft models failed to simulate growth in M9 minimal media with glucose (despite capturing ≥99% of the genes and reactions in the corresponding complete models). It is expected that all *Kp*SC models should be able to simulate growth on M9 media with glucose as a sole carbon source, as this central metabolism is universal amongst *Kp*SC. To replace missing, critical reactions required for growth on M9 with glucose, we investigated model gap-filling using the *patch_model*

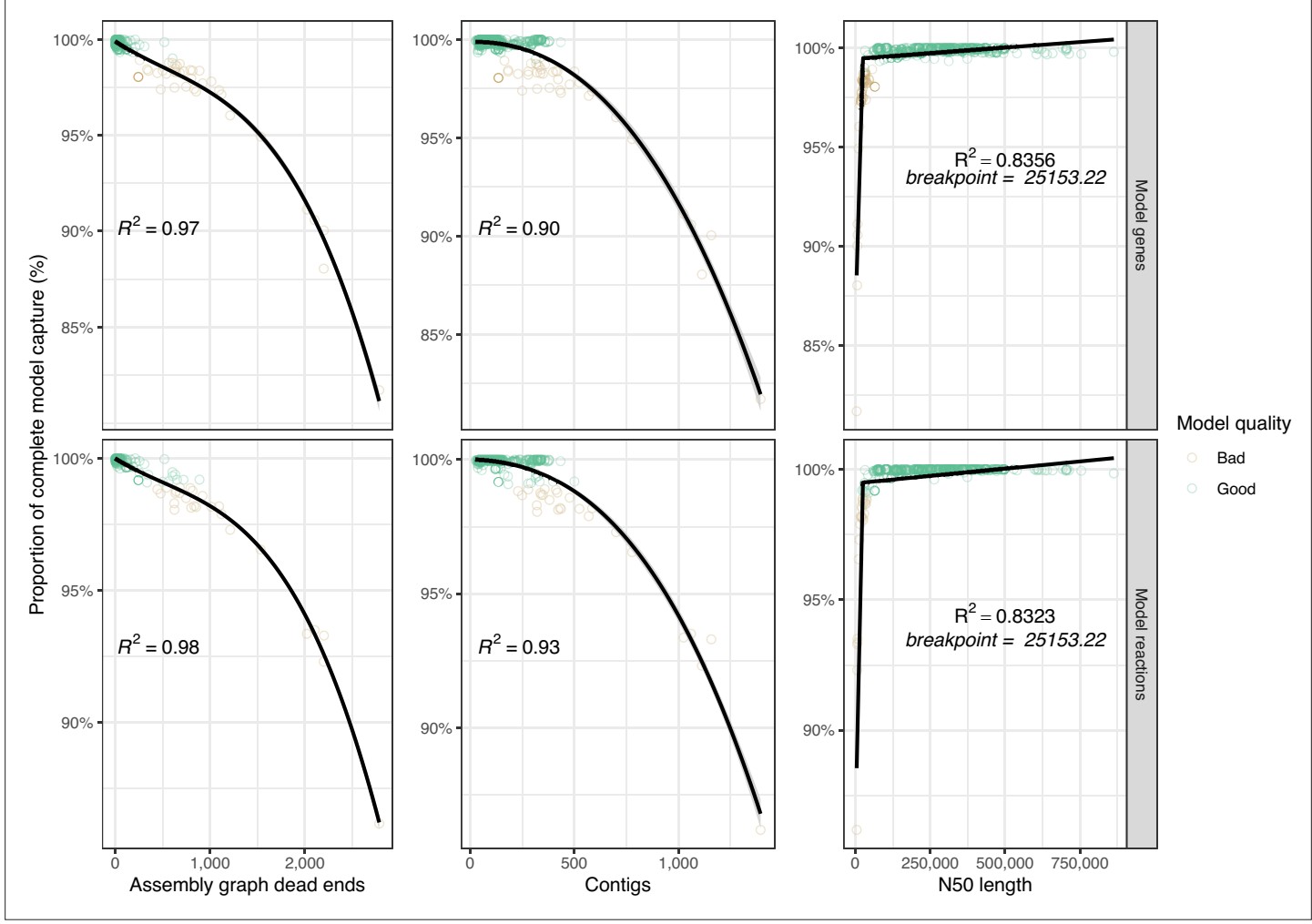

**Figure 3.** Scatterplots showing distribution of best-performing assembly metrics 'assembly graph dead-ends', 'contigs', and 'N50' against model feature capture (genes and reactions). Each point represents the mean values from a single genome (technical triplicate) and is coloured by model quality. 'Good' models capture ≥99% of the model metric as compared to the corresponding complete model (shown at each facet), 'Bad' models capture <99%. Cubic polynomial line plotted for assembly 'graph dead-ends', 'contigs', while a segmented linear model was plotted for 'N50'. $R^2$ is shown on each panel. *Figure 3—source data 1* contains additional data.

The online version of this article includes the following source data and figure supplement(s) for figure 3:

**Source data 1.** Data table showing assembly metrics and model completeness.

**Figure supplement 1.** Raincloud plot showing distributions of assembly metrics across various read subsampling depths (10× increments).

command of Bactabolize. We then assessed the accuracy of the gap-filled models for prediction of growth on the full range of substrates, as compared to the predictions from the corresponding complete models.

Gap-filling added one to three missing reactions to each model, with a median of 1, fully restoring biomass production in M9 media with glucose in all but 2 of the 23 failed models. The missing reactions appeared to be random genes across these 23 genomes, likely due to missing information in these assemblies.

Substrate usage predictions from the 21 successfully gap-filled models were highly accurate, with 18/21 having a prediction concordance of ≥99% across all 846 growth conditions (12/21 had 100% concordance) (*Supplementary file 1*). We therefore conclude that models generated for genome assemblies passing our QC criteria, which have been gap-filled to successfully simulate growth on minimal media plus glucose, are suitable for the prediction of growth across a range of substrates.

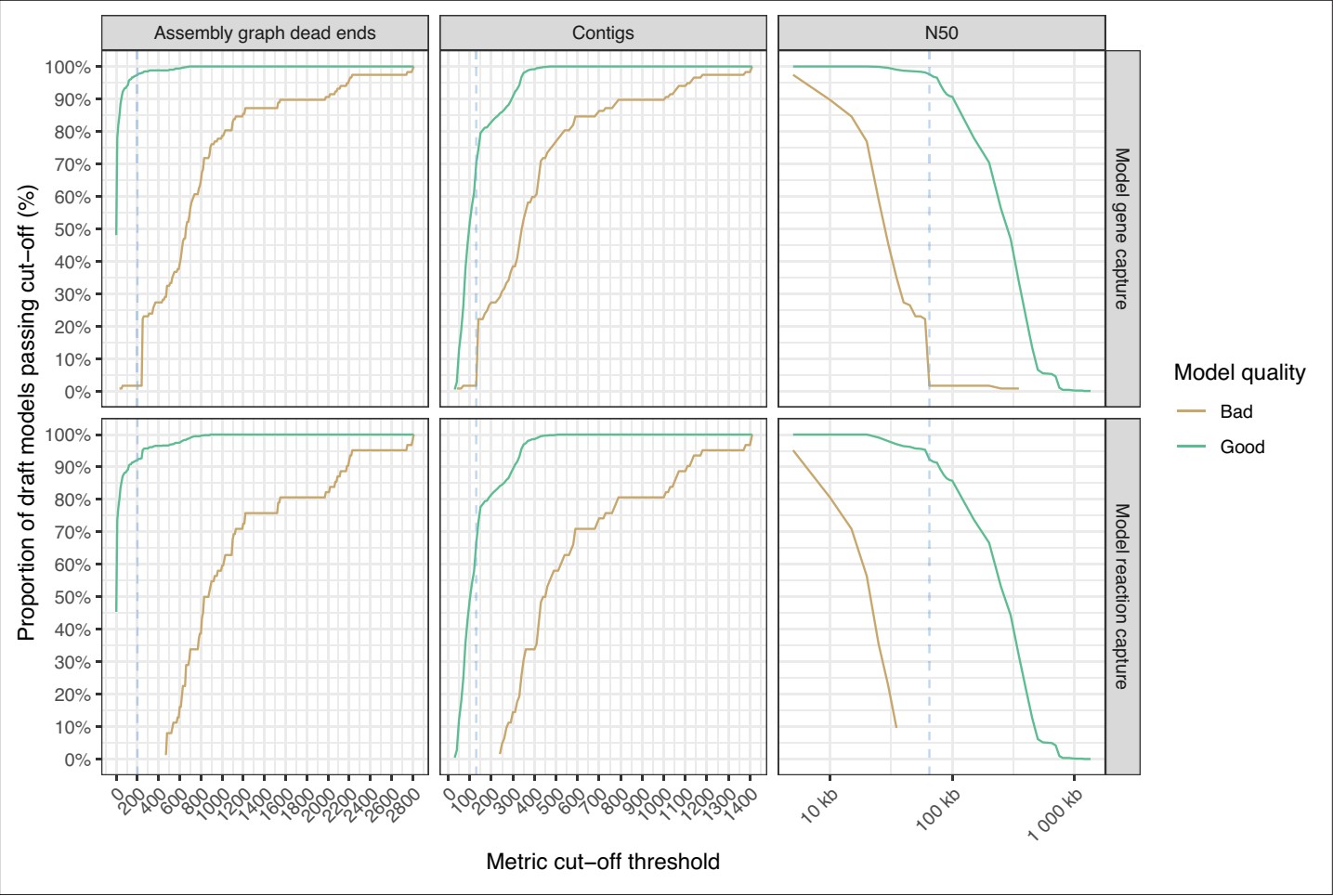

**Figure 4.** Line graphs showing the impact of assembly metric cut-off thresholds on model feature capture (n=1040). 'Good' models which captured ≥99% of model features are shown in green, while 'bad' models captured <99% model features are shown in gold. The blue dotted line shows the metric cut-off thresholds, to minimise the number of models that capture <99% model features and maximise models that capture ≥99%. Metric cut-off statistics are calculated in intervals of 10 for assembly graph dead-ends and contigs, and every 5000 for N50.

The online version of this article includes the following figure supplement(s) for figure 4:

**Figure supplement 1.** Line graph showing the capture of model features of draft assemblies (short-read only) at various depths, compared to the corresponding completed genome (long-read+short-read assemblies).

## Predictive accuracy of draft models

We assessed the accuracy of Bactabolize for the construction of draft models for 10 novel *Kp*SC clinical isolates, representing five of the major taxa in the complex. We included five isolates for which the associated STs were represented in the *Kp*SC-pan v1 model and five isolates with STs that were not represented. Whole-genome sequence data were generated on the Illumina platform and draft assemblies generated de novo. The resultant assemblies had 0–4 graph dead-ends, N50s of 15,1958–388,486 bp and 83–187 contigs (*Figure 5—source data 1*), within the tiered threshold values.

FBA was performed, and the predicted growth profiles compared to matched phenotypic growth data for 16 carbon sources derived from Vitek GN ID cards. Though the number of tested carbon sources was limited, all were associated with high accuracy metrics (*Figure 5*, *Figure 5—source data 1*). As expected, models for isolates with STs represented in the *Kp*SC-pan v1 reference performed slightly better (mean accuracy = 0.98) than those for non-represented STs (mean accuracy = 0.95).

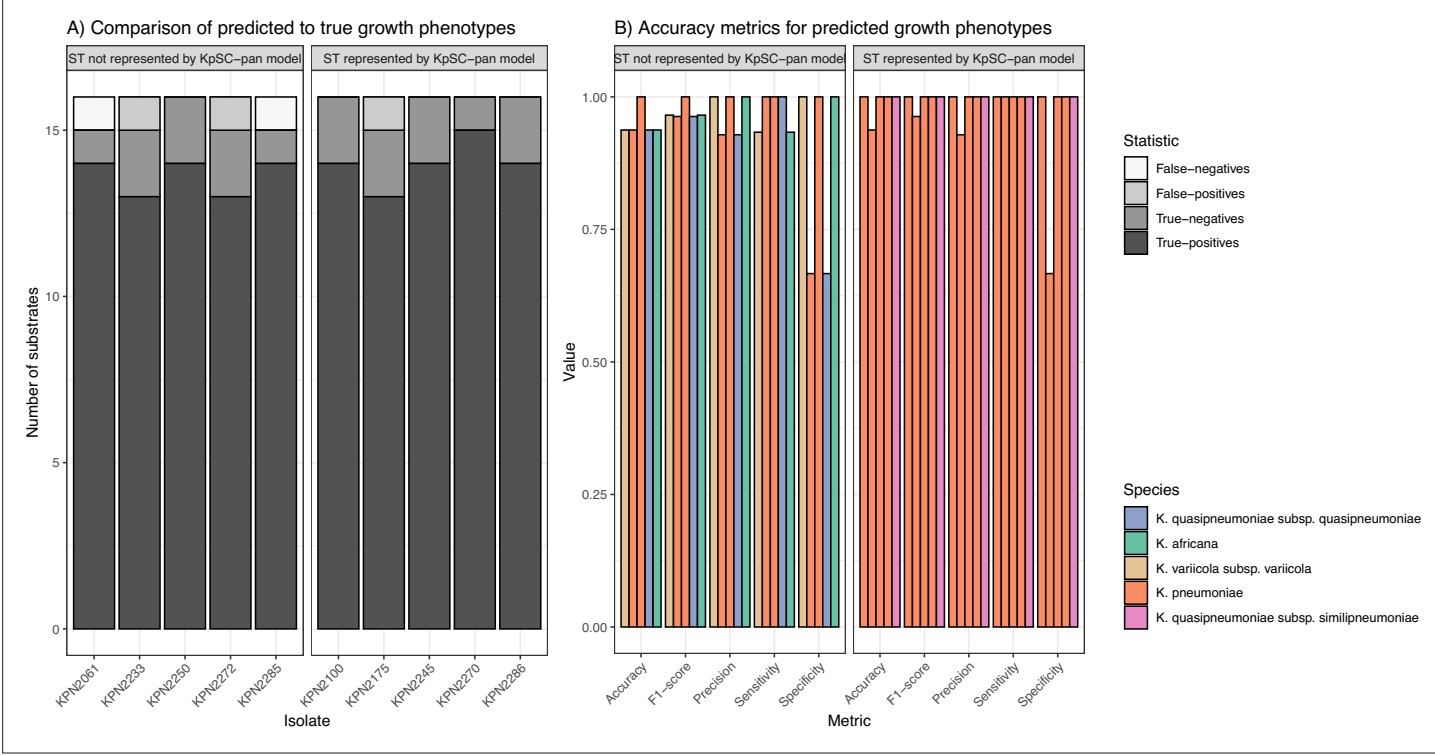

**Figure 5.** Accuracy of predicted growth phenotypes using additional isolates. (**A**) Comparisons of predicted to true phenotypes for 16 carbon source substrates. False-negatives, true-negatives, false-positives, and true-positives are coloured as shown in legend. Each column represents a different isolate, separated by ST representation in the *K. pneumoniae* species complex (*Kp*SC)-pan model. (**B**) Accuracy metrics for predicted vs phenotypic growth comparisons shown in A. Each column represents a different isolate, coloured by taxa and separated by ST representation in the *Kp*SC-pan v1 model. Additional information can be found in *Figure 5—source data 1*.

The online version of this article includes the following source data for figure 5:

**Source data 1.** Growth phenotyping results for 10 isolates used for validation.

## Discussion

In this work we described Bactabolize, a pipeline for rapid and scalable production of accurate bacterial strain-specific metabolic models and growth phenotype predictions. We describe a pan reference model for the *Kp*SC and demonstrate that a draft strain-specific model generated de novo via Bactabolize using the *Kp*SC-pan v1 reference was highly accurate for growth phenotype prediction (85.79% accuracy for substrate usage across 190 substrates, and 80.57% for gene essentiality across 1220 genes). Importantly, we also described a quality control framework for the use of draft genome assemblies as input for metabolic reconstructions. We used a systematic analysis to: (i) evaluate the proportion of gene and reaction capture compared to the corresponding 'completed' models; (ii) define quality control thresholds for input assemblies (three-tier approach for *Kp*SC; ≤200 assembly graph dead-ends, followed by ≥65,000 N50, followed by ≤130 contigs); and (iii) estimate the accuracy of the resultant growth predictions. While the quality control thresholds and accuracy estimates are specific to *Kp*SC, the conceptual framework can be applied to any organism and is essential to support the confident application of metabolic modelling for large-scale genome datasets. We appreciate that assembly graphs may not be available for dead-end count, for example for draft genome assemblies accessed via public repositories, however we encourage users to include this information in their quality control procedures wherever possible (e.g. using the recently published counter tool [*Wick, 2023*]) because these counts represent a direct reflection of the completeness of the genome assembly. In contrast, contig counts and N50 are influenced by biological features such as repeat copy numbers as well as the underlying sequence data quality, for example a bacterial genome harbouring many insertion sequence insertions will result in a draft assembly with a high number of contigs regardless of the sequence data quality and completeness.

Traditionally, genome-scale metabolic reconstruction approaches have relied upon significant manual curation efforts. While there will always remain a need for high-quality curated models, such resource-intensive approaches preclude their application at scale, and have therefore limited analyses to small numbers of individual strains (*Henry et al., 2017*; *Liao et al., 2011*). However, automated reconstruction approaches can support the generation and comparison of multiple strain-specific draft models from which meaningful biological insights can be derived (*Magnúsdóttir et al., 2017*). Additionally, the quality of curated models is likely to vary depending on their age, level, and type of curation, as well as the approach used for preliminary drafting. Indeed it is possible for automated approaches to outperform manually curated models; a draft model for *K. pneumoniae* KPPR1 generated using Bactabolize with the KpSC pan-v1 reference model outperformed the manually curated iKp1289 model representing the same strain (*Henry et al., 2017*). iKp1289 was published in 2017 (6 years prior to this study) and was initially drafted via the KBase pipeline (*Arkin et al., 2018*), which uses RAST to annotate the sequences with Enzyme Commission numbers. It has been demonstrated several times that the Enzyme Commission scheme has systematic errors (*Green and Karp, 2005*; *Rembeza and Engqvist, 2021*), leading to a loss in accuracy when compared to the ortholog identification methods used by automated approaches. Consistent with this assertion, our draft KPPR1 model constructed with KBase (without manual curation) was an outlier in terms of the very low number of genes, reactions, and metabolites that were included.

CarveMe with universal model (*Machado et al., 2018*) and gapseq (*Zimmermann et al., 2021a*) are the current gold standard automated approaches for model reconstruction, and we show that a draft *Kp*SC model generated by Bactabolize with the *Kp*SC pan v1 reference resulted in similar or better accuracy for phenotype prediction (*Figure 2*). Both the CarveMe universal and gapseq models resulted in high numbers of true-positive and true-negative growth predictions. However, these were also accompanied by comparatively higher numbers of false-positive predictions that resulted in a lower overall accuracy for substrate usage analysis compared to Bactabolize with the *Kp*SC-pan v1 reference (*Figure 2*), and comparatively lower precision and specificity for the gene essentiality analysis. False-positive predictions may indicate that the relevant metabolic machinery are present in the cell but were not active during the growth experiments (e.g. due to lack of gene expression). In this regard, false-positives are not always a sign of model inaccuracy. However, false-positive predictions can also occur from incorrect gene annotations, for example due to reduced specificity of ortholog assignment resulting from the use of the universal model without manual curation. Given a key objective here is to facilitate high-throughput analysis for large numbers of genomes, it is not feasible to expect that all models will be manually curated, and therefore we believe that identifying fewer genes with lower overall error rates provides greater confidence in the resulting draft models. We also note that the BiGG universal reference model which CarveMe leverages is no longer being actively maintained. In contrast, user-defined reference models can be iteratively curated and updated to incorporate new knowledge and data as they become available.

Bactabolize's reference-based reconstruction approach is reductive, meaning the resultant draft models will comprise only the genes, reactions, and metabolites present in the reference, or a subset thereof, and will not include novel reactions unless they are manually identified and curated by the user. This is an important caveat that should be considered carefully for application of Bactabolize to large genome datasets, particularly for genetically diverse organisms such as those in the *Kp*SC. For optimum results we suggest using a curated pan-model that captures as much diversity as possible for the target species or group of interest. While we acknowledge that a reasonable resource investment is required to generate a high-quality reference, we have shown that a pan-model derived from just 37 representative strains can be sufficient to support the generation of highly accurate draft models (*Figures 2 and 5*). Additionally, we note that it is possible to use a single strain reference model, which should ideally represent the same or closely related species to that of the input genome assemblies, in order to facilitate accurate identification of gene orthologs. It is technically possible to use an unrelated reference model, but this is expected to result in inaccurate and/or incomplete outputs and has not been tested in this study. In circumstances were no high-quality closely related reference model is available, we recommend alternative reconstruction approaches that leverage universal databases, for example CarveMe (*Machado et al., 2018*) or gapseq (*Zimmermann et al., 2021a*). However, gapseq's long compute time makes it inappropriate for application to datasets comprising 100s–1000s of genomes (such as have become increasingly

common in the bacterial population biology literature) (*Ludden et al., 2020*; *Dyson and Holt, 2021*).

Bactabolize and the *Kp*SC pan v1 model are freely available under open source licenses and satisfy the four features of the FAIR research principles (findability, accessibility, interoperability, and reusability) (*Wilkinson et al., 2016*). In addition to the *Kp*SC pan reference described here, a pan reference model has been described previously for *S. enterica* (representing 410 strains) (*Seif et al., 2018*), *Bacillus subtilis* (*Blázquez et al., 2023*) (representing 183 strains), and *E. coli* (*Monk, 2022*) (representing 222 strains). We are actively working to expand and improve the *Kp*SC pan reference model and welcome similar efforts to generate high-quality references for other organisms. Together these resources will facilitate population-wide metabolic analyses for global priority pathogens, which can be used to understand how they transmit, cause disease, and evolve drug resistance, and to identify novel therapeutic targets.

# Materials and methods

## Key resources table

| Reagent type (species) or resource | Designation | Source or reference | Identifiers | Additional information |
|---|---|---|---|---|
| Software, algorithm | Bactabolize | This study – Data availability section | | Software |
| Software, algorithm | Cobrapy | *Ebrahim et al., 2013* | | Software |
| Strain, strain background | *K. pneumoniae* KPN2061, KPN2100, KPN2175, KPN2233, KPN2245, KPN2250, KPN2270, KPN2272, KPN2285, KPN2286 | This study – See *Source data 1* and *Figure 5—source data 1* for details and accessions | | Isolates used to validate Bactabolize |
| Commercial assay or kit | VITEK 2 GN ID cards | bioMérieux | | Method used to validate Bactabolize |

## Bactabolize pipeline

Bactabolize utilises the existing metabolic modelling library COBRApy (*Ebrahim et al., 2013*) and Python 3 (*Van Rossum and Drake, 2009*). All code is freely available and open source at GitHub (*Watts et al., 2023*) under a GNU General Public License v3.0. Users should additionally cite COBRApy (*Ebrahim et al., 2013*) if Bactabolize is used.

## *KpSC* pan-metabolic model

The 37 metabolic models from a previous study (*Hawkey et al., 2022*) were combined with the iY1228 model using the create_master_model.py script (available at 10.6084/m9.figshare.21728717). Briefly, all GPRs from the iYL1228 model and the associated sequences were included, as well as new GPRs identified from the 36 additional strains by manual curation following comparison to the matched phenotype data (as described in *Hawkey et al., 2022*). Additionally, orthologous sequence variants with <75% nucleotide identity to gene sequences associated with these GPRs were added if there was phenotype data supporting the reaction. The biomass reaction was updated, removing the metabolites udpgalur_c and udpgal_c as their production was strain-specific.

Metadata annotations were improved using the improve_model_annotations.py script (also available in the Bactabolize code repository) resulting in the KpSC_pan v1 used in this study (*Vezina et al., 2023*).

## Draft model generation

The annotated and unannotated genome of *K. pneumoniae* KPPR1 were obtained from Genbank accession number: CP009208.

Bactabolize draft models were generated using the *draft_model* command in Bactabolize v1 with the *Kp*SC-pan v1 model as a reference, the annotated *K. pneumoniae* KPPR1 as input and the following options:

```
-min_coverage 25 --min_pident 80 --media M9 --atmosphere aerobic
```

A draft model for *K. pneumoniae* KPPR1 was also generated via CarveMe version 1.5.1 using the universal reference, the annotated *K. pneumoniae* KPPR1 as input, with the following commands: '-g

M9 -i M9'. Subsequently, the `--universe-file` mode was also used, so the *Kp*SC-pan model could be used as a reference, with the previously described command.

A draft model was generated using gapseq version 1.2 with the 'doall' command using the unannotated genome (as gapseq does not take annotated input files). Gap-filling was subsequently performed using the 'fill' command and a custom M9 media file to match the nutrient list found in Bactabolize. Finally, a draft model was constructed using the annotated genbank *K. pneumoniae* KPPR1 file and the KBase narrative (***Henry et al., 2017***).

## Speed calculations

Modelling methods were timed via a script using the date +%s.%N command run before and after command on the MASSIVE computing cluster (Intel Xeon Gold 6150 CPU @ 2.70 GHz and 155 GB of memory, CentOS Linux release 7.9.2009 environment). 10 individual complete *Kp*SC genomes used in the quality control framework were tested for each method and the mean and range reported using R version 4.0.3 (***R Development Core Team, 2020***).

## Performance comparisons

The annotated and unannotated genomes of *K. pneumoniae* KPPR1 were obtained from Genbank under the accession: CP009208, and draft metabolic models were generated using Bactabolize, CarveMe, ModelSEED, and gapseq as described above. The previously described, manually curated model for KPPR1 (iKp1289) was also included for comparison (***Henry et al., 2017***). MEMOTE version 0.13.0 was used to collate basic model statistics. The following KPPR1 phenotype data were retrieved from published studies: BIOLOG Phenotypic Microarray data (***Henry et al., 2017***) and single-gene knockout data inferred from the outputs of a TraDIS transposon mutagenesis library (***Short et al., 2020***).

A list of BIOLOG growth substrates for plates PM1, PM2A, PM3B, and PM4A (***Biolog, 2020***) were converted where possible to BiGG and SEED IDs by manual search of the BiGG (bigg.ucsd.edu) and SEED websites (https://modelseed.org/biochem/compounds). An updated BiGG to SEED dictionary can be found in ***Source data 1***. A total of 143 of 190 carbon, 82 of 95 nitrogen, 46 of 59 phosphor, and 26 of 35 sulphur substrates were successfully matched to BiGG and SEED IDs (***Source data 1***). These growth data were compared to in silico predictions generated via FBA using the *fba* command from Bactabolize to optimise the biomass objective function with the following options:

`–fba_spec_name m9 --fba_open_value –20`

Gene essentiality was inferred from single-gene knockout growth predictions using the *sgk* command from Bactabolize with the following options to mirror the growth conditions of the TraDIS library (LB media grown aerobically):

`-media_type lb --atmosphere aerobic`

In all cases, an objective value cut-off of ≥10⁻⁴ was used to indicate binarised growth as per previous studies (***Hawkey et al., 2022***; ***Norsigian et al., 2019***).

In silico predictions were compared to matched phenotype data and the following accuracy metrics were calculated:

$$Precision = \frac{TP}{TP + FP}$$

$$Sensitivity/recall = \frac{TP}{TP + FN}$$

$$Specificity = \frac{TN}{TN + FP}$$

$$Accuracy = \frac{TP + TN}{TP + FP + TN + FN}$$

$$F1 - score = 2 \times \frac{Precision \times Sensitivity}{Precision + Sensitivity}$$

Model metabolite and reaction IDs were harmonised for overlap comparisons using the 'Import model' function from MetaNetX.org (***Moretti et al., 2021***) after import and export via the write_sbml_model function from COBRApy.

### Quality control framework

Illumina read sets (250 bp paired end) and completed genome sequences for 37 *Kp*SC isolates were described previously (*Hawkey et al., 2022*). Here, we randomly subsampled the Illumina reads at various depths (10–100, by increments of 10) using rasusa version 0.3.0 (*Hall, 2019*) in technical triplicate. Reads were then trimmed using TrimGalore version 0.5.0 (*Krueger, 2012*) and assembled de novo with Unicycler version 0.4.7 (*Wick et al., 2017*), default parameters. Assembly statistics and assembly graph dead-ends were calculated using the GFA-dead-end-counter version 1.0.0 (*Wick, 2023*). Draft metabolic models were generated with Bactabolize using the *Kp*SC-pan v1 reference, and growth substrate profiles were predicted as described above. We compared the outputs from models generated for draft genome assemblies to those generated for the corresponding completed genomes. Where necessary models were gap-filled via the *patch_model* command.

### Predictive accuracy of draft models

Novel growth phenotype data were generated for 10 *Kp*SC clinical isolates from our in-house collection using the VITEK 2 GN ID card system as described previously (*Hawkey et al., 2022*). Briefly, isolates were grown on Tryptic Soy (OXOID) agar plates overnight at 37°C, then analysed using VITEK 2 GN ID cards (bioMérieux) and read on the VITEK 2 Compact (bioMérieux) as per the manufacturer's instructions using software version 8.0. DNA was extracted for whole-genome sequencing via Genfind v3 extraction kit, library preparation performed using Nextera Flex (Illumina) using ¼ reagents. Paired-end read data (300 bp) were generated on an Illumina NovaSeq6000 SP v1.0 and have been deposited in the European Nucleotide Archive under Bioproject PRJNA777643 (individual read accession numbers are given in *Figure 5—source data 1*). Draft genome assemblies were generated with Unicycler, and draft metabolic models and growth predictions were generated with Bactabolize as described above.

### Statistics and visualisation

Statistical analysis and graphical visualisation were performed using R version 4.0.3 (*R Development Core Team, 2020*), RStudio version 1.3.1093 (*RStudio-Team, 2020*), with the following software packages: tidyverse version 1.3.1 (*Wickham et al., 2019*), viridis version 0.5.1 (*Garnier, 2018*), RColorBrewer version 1.1–2 (*Neuwirth, 2022*), ggpubr version 0.4.0 (*Kassambara, 2023*), ggpmisc version 0.4.4 (*Aphalo et al., 2023*), aplot version 0.1.6 (*Yu, 2023*), colorspace version 2.0–2 (*Zeileis et al., 2020*), ggpattern version 0.4.3–3 (*Mike and Davis, 2022*), ggtext version 0.1.1 (*Wilke and Wiernik, 2020*), and glue version 1.4.2 (*Hester, 2022*).

Linear regression analysis was performed in R using the lm function and a third-degree polynomial model was fitted to plots with the following equation: y~poly(x, 3, raw = TRUE). The segmented linear model was fitted using segmented version 1.6–2 (*Muggeo VMR, 2023*).

All code used to generate results can be found as supplemental material (*Watts et al., 2023* and *Vezina et al., 2023*) and on Figshare.

### Logo

The Bactabolize logo was constructed in Inkscape version 1.0.1 (*Inkscape, 2020*). The font used is Proportional TFB (*zanatlija, 2012*) and Element (*Weknow, 2015*).

## Acknowledgements

We thank Sylvain Brisse for the isolates used in this study and review of the manuscript. Funding. This work was funded by the Australian Research Council Discovery Project DP200103364. KLW is supported by the National Health and Medical Research Council of Australia (Investigator Grant APP1176192).

# Additional information

## Funding

| Funder | Grant reference number | Author |
| --- | --- | --- |
| Australian Research Council | DP200103364 | Kelly L Wyres |
| National Health and Medical Research Council | APP1176192 | Kelly L Wyres |

The funders had no role in study design, data collection and interpretation, or the decision to submit the work for publication.

## Author contributions

Ben Vezina, Conceptualization, Software, Formal analysis, Investigation, Visualization, Methodology, Writing – original draft, Writing – review and editing; Stephen C Watts, Software, Methodology, Writing – review and editing; Jane Hawkey, Conceptualization, Methodology; Helena B Cooper, Validation, Writing – review and editing; Louise M Judd, Methodology, Writing – review and editing; Adam WJ Jenney, Resources, Writing – review and editing; Jonathan M Monk, Conceptualization, Supervision, Funding acquisition, Methodology; Kathryn E Holt, Supervision, Funding acquisition, Writing – review and editing; Kelly L Wyres, Conceptualization, Formal analysis, Supervision, Funding acquisition, Validation, Investigation, Methodology, Writing – original draft, Project administration, Writing – review and editing

## Author ORCIDs

Ben Vezina ⓘ http://orcid.org/0000-0003-4224-2537
Helena B Cooper ⓘ http://orcid.org/0000-0002-7096-0010
Kathryn E Holt ⓘ http://orcid.org/0000-0003-3949-2471
Kelly L Wyres ⓘ http://orcid.org/0000-0002-4033-6639

Reviewer #1 (Public Review): https://doi.org/10.7554/eLife.87406.3.sa1
Reviewer #2 (Public Review): https://doi.org/10.7554/eLife.87406.3.sa2
Author Response https://doi.org/10.7554/eLife.87406.3.sa3

---

# Additional files

## Supplementary files

• Supplementary file 1. Faceted graphs showing the number of substrate usage (fba module) discrepancies of gap-filled models (patch_model module) which initially did not produce biomass (models which failed to simulate growth). The dots indicate percentage concordance with the completed genome model, while the columns indicate number of substrates with discrepancies (no simulated growth in patched model, but growth in completed genome model).

• MDAR checklist

• Source data 1. Table showing all BIOLOG phenotypic growth data.

## Data availability

All genomes used are publicly available and generated by referenced studies. Source data 1 and Figure 5-source data 1 shows all accession numbers. All supporting data are included in Figure 2-source data 1, Figure 3-source data 1, Figure 5-source data 1 and Source data file 1. All Bactabolize code and data generated for this study can be found on GitHub (copy archived at *Watts et al., 2023*). The KpSC-pan v1 reference model is freely available on GitHub (copy archived at *Vezina et al., 2023*). Additional data is available on Figshare.

The following dataset was generated:

| Author(s) | Year | Dataset title | Dataset URL | Database and Identifier |
|---|---|---|---|---|
| Vezina B, Watts S, Hawkey J, Cooper H, Judd L, Jenney A, Monk J, Holt KE, Wyres K | 2023 | Bactabolize additional resources | https://doi.org/10.6084/m9.figshare.21728717.v2 | figshare, 10.6084/m9.figshare.21728717 |

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

github.com/jotech/gapseq/issues/77

