## [Editor Report · eLife assessment]

This study presents Bactabolize, a **valuable** tool for the rapid genome-scale reconstruction of bacteria and the prediction of growth phenotypes. Using validated methodology, the tool relies on a reference pan-genome model to create strain-specific draft metabolic models, as demonstrated in this study using *Klebsiella pneumoniae*. While the evidence in this specific case is **solid**, validation across diverse bacterial species is yet to be confirmed.

---

## [Referee Report · Reviewer #1 (Public Review)]

In this work, Vezina et al. present Bactabolize, a rapid reconstruction tool for the generation of strain-specific metabolic models. Similar to other reconstruction pipelines such as CarveMe, Bactabolize builds a strain-specific draft reconstruction and subsequently gap-fills it. The model can afterwards be used to predict growth on carbon sources. The authors constructed a pan-model of the Klebsiella pneumoniae species complex (KpSC) and used it as input for Bactabolize to construct a genome-sale reconstruction of K. pneumoniae KPPR1. They compared the generated reconstruction with a reconstruction built through CarveMe as well as a manually curated reconstruction for the same strain. They then compared predictions of carbon, nitrogen, phosphor, and sulfur sources and found that the Bactabolize reconstruction had the overall highest accuracy. Finally, they built draft reconstructions for 10 clinical isolates of K. pneumoniae and evaluated their predictive performance. Overall, this is a useful tool, the data is well-presented, and the paper is well-written.

---

## [Referee Report · Reviewer #2 (Public Review)]

The authors present a pipeline for generating strain-specific genome-scale metabolic models for bacteria using Klebsiella spp. as the demonstrative data. This paper claims to provide a high-throughput tool for generating strain-specific models for bacteria. However, in reality, the tool requires a reference pan-genome-based complete model to generate the strain-specific model of the species of interest, which in this study is Klebsiella pneumoniae. This requirement renders the tool redundant for high-throughput purposes since the process of building or generating the pan-genome reference model is performed separately. Additionally, the quality of the newly built strain-specific model will depend on the reference model used. Therefore, this tool, on its own, can only work specifically with the available pan-genome model of reference, which in this case is only applicable to Klebsiella pneumoniae. Its effectiveness with other bacteria has not been proven. I would suggest that the authors either reframe the performance and results to be applicable only to Klebsiella or consider adding more reference pan-genome models for the study.

---

## [Author Response]

The following is the authors’ response to the original reviews.

**Reviewer #1**
1. Overall, this is a useful tool, the data is well-presented, and the paper is well-written. However, the predictions are only compared with two existing reconstruction tools though more have been recently published

The aim of this work was to facilitate high-throughput generation of strain-specific metabolic models e.g. at the scale of 100s -1000s as indicated throughput the introduction (see lines 74-82, 91-94), and therefore we only compared tools which were capable of high-throughput analysis via command line and excluded others (e.g. merlin). We have now tested this against the other recent command line tool, gapseq which had escaped our gaze. Thank you for bringing this to our attention. Additionally, we have included KBase (ModelSEED, a web-based app that does not support high-throughput analysis) to allow for readers to interpret the results in the context of community standard approaches, since KBase is a popular tool.

We have added an explicit statement about the choice of approaches, now at lines 194-199 as follows:

“We compared the output and performance of Bactabolize to the two previously published tools that can support high-throughput analyses i.e. CarveMe (30) and gapseq (31). To aid interpretation in the context of community standard approaches, we also include a comparison to the popular web-based reconstruction tool, KBase (ModelSEED), and a manually curated metabolic reconstruction of K. pneumoniae strain KPPR1 (also known as VK055 and ATCC 43816, metabolic model named iKp1289) (15).”

The methods section was updated accordingly (now lines 552-558):

“A draft model was generated using gapseq version 1.2 with the ‘doall’ command using the unannotated genome (as gapseq does not take annotated input files). Gap-filling was subsequently performed using the ‘fill’ command and a custom M9 media file to match the nutrient list found inBactabolize(https://github.com/kelwyres/Bactabolize/blob/main/data/media_definitions/m9 _media.json).

Finally, a draft model was constructed using the annotated genbank K.pneumoniae KPPR1 file and the KBase narrative (15) (https://narrative.kbase.us/narrative/ws.14145.obj.1).”

The results section (now lines 193-308) and Figure 2 have also been updated / restructured to reflect the new analyses, and include a comparison of the relative compute times for the construction of models (lines 281-291) as follows:

“While model features and accuracy are essential metrics for comparison, computation time is also a key consideration for high-throughput analyses. We recorded the time required for each tool to build draft models for 10 of the completed KpSC genomes used in the quality control framework (see below) on a high-performance computing cluster Intel Xeon Gold 6150 CPU @ 2.70GHz and 155 GB of requested memory on a CentOS Linux release 7.9.2009 environment. CarveMe KpSC pan was the fastest with a mean of 20.04 (range 19.90 - 20.18) seconds, followed by CarveMe universal at 30.28 (range 29.20 - 31.80) seconds, then Bactabolize KpSC pan at 98.05 (range 92.19 - 100.4) seconds. KBase took 183.50 (range 120.00 - 338.00) seconds per genome via batch analysis, including genome upload time and queuing. gapseq took 5.46 (range 4.55 - 6.28) hours to produce draft models (not including the required gap-filling), consistent with previous reports (37).”

Finally, the whole discussion has been updated and substantially restructured (lines 472-474, 475-493, 494-512). Specific mentions to the new analyses are at lines:

472-474: “Consistent with this assertion, our draft KPPR1 model constructed with KBase (without manual curation) was an outlier in terms of the very low number of genes, reactions and metabolites that were included.”

475-493: “CarveMe with universal model (30) and gapseq (31) are the current gold standard automated approaches for model reconstruction, and we show that a draft KpSC model generated by Bactabolize with the KpSC pan v1 reference resulted in similar or better accuracy for phenotype prediction (Figure 2). Both the CarveMe universal and gapseq models resulted in high numbers of true-positive and true-negative growth predictions. However, these were also accompanied by comparatively higher numbers of false-positive predictions that resulted in a lower overall accuracy for substrate usage analysis compared to Bactabolize with the KpSC-pan v1 reference (Figure 2), and comparatively lower precision and specificity for the gene essentiality analysis. False-positive predictions may indicate that the relevant metabolic machinery are present in the cell but were not active during the growth experiments (e.g. due to lack of gene expression). In this regard, false-positives are not always a sign of model inaccuracy. However, false-positive predictions can also occur from incorrect gene annotations e.g. due to reduced specificity of ortholog assignment resulting from the use of the universal model without manual curation. Given a key objective here is to facilitate high-throughput analysis for large numbers of genomes, it is not feasible to expect that all models will be manually curated, and therefore we believe that identifying fewer genes with lower overall error rates provides greater confidence in the resulting draft models. We also note that the BiGG universal reference model which CarveMe leverages is no longer being actively maintained. In contrast, user defined reference models can be iteratively curated and updated to incorporate new knowledge and data as they become available.”

510-512: “However, gapseq’s long compute time makes it inappropriate for application to datasets comprising 100s-1000s of genomes (such as have become increasingly common in the bacterial population biology literature).”

1. My understanding is that the tool requires a set of reference reconstructions for other strains of the target species. If no reference reconstruction is available for another strain of the target species, can this species not be reconstructed?

Any input reference can be used to generate models however, single strain models matching the target species, or ideally a species-specific panreference, are recommended for best results. We have added a discussion on these points at lines 128-133:

“For optimum results we suggest using a pan-model that captures as much diversity as possible for the target species or group of interest, because Bactabolize’s reconstruction method is reductive i.e. each output strainspecific model will include only genes, reactions and metabolites that are present in the reference or a subset thereof (although novel genes, reactions and metabolites can be added via manual curation).”

We expand on these points further in the discussion:

494-512: “Bactabolize’s reference-based reconstruction approach is reductive, meaning the resultant draft models will comprise only the genes, reactions and metabolites present in the reference, or a subset thereof, and will not include novel reactions unless they are manually identified and curated by the user. This is an important caveat that should be considered carefully for application of Bactabolize to large genome data sets, particularly for genetically diverse organisms such as those in the KpSC. For optimum results we suggest using a curated pan-model that captures as much diversity as possible for the target species or group of interest. While we acknowledge that a reasonable resource investment is required to generate a high-quality reference, we have shown that a pan-model derived from just 37 representative strains can be sufficient to support the generation of highlyaccurate draft models (Figure 2 and 5). Additionally, we note that it is possible to use a single strain reference model, which should ideally represent the same or closely related species to that of the input genome assemblies, in order to facilitate accurate identification of gene orthologs. It is technically possible to use an unrelated reference model, but this is expected to result in inaccurate and/or incomplete outputs and has not been tested in this study. In circumstances were no high quality closely-related reference model is available, we recommend alternative reconstruction approaches that leverage universal databases e.g. CarveMe (30) or gapseq (31). However, gapseq’s long compute time makes it inappropriate for application to datasets comprising 100s-1000s of genomes (such as have become increasingly common in the bacterial population biology literature).”

1. How do the reconstructions generated by Bactabolize compare to those generated by other reconstruction tools besides CarveMe and ModelSEED, e.g., gapseq (Zimmermann et al, Genome Biology 2021. 22:81) or merlin Capela et al, Nucleic Acids Res 2022, 50(11):6052-6066?

See response to rev 1 point 1.

1. How are the accuracy, specificity, and sensitivity of the pan-models calculated? Is the compared experimental data on the species level?

We used the pan-model as a reference from which we generated a strain-specific model for K. pneumoniae KPPR1 (using Bactabolize and CarveMe). This strain-specific metabolic model was then used to simulate growth phenotypes and compared to published experimental data for KPPR1. This was described in the methods section, including the calculations for themetrics (lines 589-593); however, we have also expanded the description within the results section to clarify the approach (lines 201-209):

“De novo draft models for strain KPPR1 were built using; (i) Bactabolize with the KpSC pan v1 reference; (ii) CarveMe, with its universal reference model (CarveMe universal); (iii) CarveMe, with KpSC-pan v1 reference (CarveMe KpSC pan); (iv) gapseq; and (v) KBase (ModelSEED). ….. Subsequently, each model was used to predict growth phenotypes; (i) in M9 minimal media with different sole sources of carbon, nitrogen, phosphorus and sulfur; and (ii) for all possible single gene knockouts in LB under aerobic conditions. The predicted phenotypes were compared directly to the published phenotype data.” [Note the published data are cited in the previous manuscript sentence, not shown here].

1. The link https://github.com/rrwick/GFA-dead-end-counter, in line 286 does not work.

Link regenerated – now at line 451-452 and 604

**Reviewer #2**
1. KpSC pan-metabolic reference model is provided. Are they required as input for Bactabolize? Are the gene, metabolite information open accessible by users? o See response to reviewer 1 point 2 above and;

All data for the KpSC pan-model described in this work are accessible in the model files and amino acid + nucleotide files + data table athttps://github.com/kelwyres/KpSC-pan-metabolic-model. This is also linked in the manuscript at line 631 and in the Data availability statement at line 661.

1. In the results section "description of Bactabolize", the authors present technical details on how to generate a metabolic model. For the input and output, please provide concrete examples to show the functionality of Bactabolize.

Detailed instructions, example code and example input/output files are available via the Bactabolize GitHub repository: https://github.com/kelwyres/Bactabolize.https://github.com/kelwyres/Bactabolize/tree/main/data/test_data

Instructions and example code can be found on the wiki:https://github.com/kelwyres/Bactabolize/wiki Test data and example files are at:

The Github repository is linked in the manuscript at lines 95, 124, 552, and 667, and we have added a further reference at line 124, which mentions the example code/data:“Full documentation, including example code and test data are available at the Bactabolize code repository (https://github.com/kelwyres/Bactabolize).”

1. To generate metabolic models, the authors present comparison results with other methods. However, the authors only present the numbers in genes, metabolites and substrates. Since the interactions between gene, metabolite, and substrate are also critical, if possible, please provide the coverage details about these interactions. Venn diagram is recommended to compare these coverage differences.

Two additional supplementary figures have been generated (Figures S5 and 6) showing Venn diagrams of metabolites and reactions for the highthroughput analysis approaches that are most relevant to this work (see also response to rev 1, point 1). These are discussed at lines 224-237:

“Figures S5 and S6 show the overlaps of metabolites and reactions between the high-throughput reconstruction methods after processing with MetaNetX (59) to standardise the reaction and metabolite nomenclatures (excluding CarveMe pan for simplicity and given the likely problems of reaction oversubscription). The majority of the reactions included in the Bactabolize model were conserved in either the CarveMe universal model (n = 1225, 53.2%), gapseq model (n = 54, 2.3%) or both (n = 665, 28.9%). The reaction overlap was skewed to the CarveMe universal model which shared 1225 reactions that were conserved in the Bactabolize model but absent from the gapseq model. Notably, the gapseq model contained a large number (2200) of unique reactions (70.4% of those in the model). Similarly, the vast majority of metabolites in the Bactabolize model were conserved in one or both of the other models (n = 917, 85.6%). However, it is likely that true overlaps between methods are underrepresented due to the different reaction identifiers and chemical synonyms used within the BiGG (Bactabolize, CarveMe) vs ModelSEED nomenclatures (gapseq), which are difficult to harmonise in an automated manner even after the application of MetaNetX.”

Figure 2 shows not only the model numbers but also includes benchmarking to real phenotypic data in 2DEFG as the key mode of comparison between models. This encompasses meaningful interactions between gene, metabolic and substrate. The results are discussed at length in text at lines 253-271:

“We assessed the performance of each model for in silico prediction of growth phenotypes compared to the previously published experimental data (15). Accuracy, sensitivity, specificity, precision and F1 scores were calculated (60). Note that the specific set of growth substrates and gene knockouts that can be simulated is determined by the sets of genes and metabolites captured by each model and is therefore model-dependent (Data S1 and S2). Among those with matched experimental phenotype data, the Bactabolize and CarveMe universal models were able to predict growth for a greater number of carbon, nitrogen, phosphorous and sulfur substrates than gapseq, CarveMe KpSC pan, KBase and iKp1289 models (Figure 2C, Data S1). While the CarveMe universal model had the highest number of truepositive growth predictions overall (n = 132 of 617 total predictions), it also had a comparably high number of false-positive predictions (n = 39 of 617 total predictions, Figure 2D). Similarly, the gapseq and iKp1289 models resulted in 31 (262 total predictions) and 50 (513 total predictions) falsepositive predictions, respectively. In contrast, the Bactabolize model had fewer false-positive predictions (n = 21 of 505 total predictions) alongside a high number of true-positive predictions (n = 117 of 505 total predictions), resulting in the highest overall accuracy metrics (Figure 2E, Data S1). The KBase model was a notable outlier, associated with a high number of falsenegative predictions (n = 31 of 103 total predictions) and low false-positive predictions (n = 3 of 103 total predictions), presumably resulting from the very low number of genes and reactions included in the model, driving low sensitivity and accuracy.”

Lines 272-280:

“The gene essentiality results showed that gapseq produced the highest absolute number of true-positive gene essentiality predictions (n = 79 of), followed by Bactabolize KpSC pan (n = 44 of 1220 total predictions), then CarveMe universal (n = 39 of 1951 total predictions). CarveMe universal had the largest number of true-negatives by a wide margin (n = 1599 of 1951 total predictions), followed by gapseq (n = 1085 of 1403 total predictions), then Bactabolize KpSC pan (n = 939 of 1220 total predictions), driving their high accuracies (83.96%, 82.96% and 80.57%, respectively). The Bactabolize model was associated with the greatest overall precision and specificity (Figures 2F & 2G) while the gapseq model resulted in the highest F1-score and sensitivity.”

1. Are quality control and gap-filling needed to be processed when constructing a new metabolic model?

Our goal here was to implement an approach to support high-throughput analyses (see response to rev 1 point 1), including leveraging draft genome assemblies as the bases for the construction of strain-specific metabolic models. As part of this work, we have described a robust quality control (QC) framework for screening draft K. pneumoniae genomes i.e. to identify genome assemblies that should not be used. We developed this framework by comparison to models generated for matched completed genomes. Our analyses demonstrate the importance of applying QC to the input draft genome assemblies. When appropriate QC is applied to the input genomes, the resultant draft models show a high degree of completeness compared to the matched models derived from complete genomes. The draft models can also be used to simulate growth phenotypes with high accuracy as compared to those simulated for the matched complete genome models.

No specific QC was applied to the draft models themselves, other than confirmation of positive growth prediction in m9 minimal media plus glucose (which is expected to support growth of all K. pneumoniae). In cases where the input assembly passed our QC criteria but the resultant model was unable to simulate growth in m9 minimal media plus glucose, gap-filling may be optionally applied. Again, by comparison to the simulated phenotypes from matched complete genome models, we show that these gap-filled draft models can produce accurate phenotype predictions. See lines 396-404:

“Of the 901 draft genome assemblies which passed our QC criteria (≤200 assembly graph dead ends), 23 of the resulting draft models failed to simulate growth in M9 minimal media with glucose (despite capturing ≥99% of the genes and reactions in the corresponding complete models). It is expected that all KpSC models should be able to simulate growth on M9 media with glucose as a sole carbon source, as this central metabolism is universal amongst KpSC. To replace missing, critical reactions required for growth on M9 with glucose, we investigated model gap-filling using the patch_model command of Bactabolize. We then assessed the accuracy of the gap-filled models for prediction of growth on the full range of substrates, as compared to the predictions from the corresponding complete models.” Lines 409-413: “Substrate usage predictions from the 21 successfully gap-filled models were highly accurate, with 18/21 having a prediction concordance of ≥99% across all 846 growth conditions (12/21 had 100% concordance) (Figure S9). We therefore conclude that models generated for genome assemblies passing our QC criteria, which have been gap-filled to successfully simulate growth on minimal media plus glucose, are suitable for the prediction of growth across a range of substrates.”

1. Are there any visualization results to check the status of the generated draft model?

No. This is a tool for large-scale and rapid production of metabolic models, and phenotype prediction and we have not included visualisation tools. Third party tools are available e.g. https://fluxer.umbc.edu/. We do provide optional generation of MEMOTE reports at lines 136-138:

“Draft genome-scale metabolic models are output in both SMBL v3.1 (41) and JSON formats (one pair of files for each independent strain-specific model), along with an optional MEMOTE quality report (42)”.

**Reviewer #3**
1. The justification and evaluation of the generated models are inadequate and onedimensional. The authors only focus on statistics such as the number of reactions and genes in the models, which does not accurately depict the completeness of the model.

The reviewer has misunderstood how we have used ‘completeness’ in this manuscript. In the section describing our novel QC framework, we use this term to refer to the relative completeness of draft models generated from draft genome assemblies as compared to curated models generated from complete genome assemblies for the same strains. The latter were considered as the ‘complete’ models for this purpose. We are not referring to any measure of network or metabolic pathway completeness. We specifically refer to gene and reaction capture compared to the ‘complete’ models because these features directly reflect the problem we are trying to addressi.e. that draft genome assemblies may not contain the complete set of genes that are truly present in the underlying genome. We have updated the manuscript text to further clarify the problem we aim to address in this section and justify the use of gene and reaction capture metrics:

Lines 310-319: “There are now thousands of bacterial genomes available in public databases, the majority of which are in draft form, comprising 10s to 1000s of assembly contigs. This fragmentation of the genome is caused by repetitive sequences that cannot be resolved by the assembly algorithm and/or sequence drop-out. The latter can result in the loss of genetic information such that some portion of genes present in the underlying genome are lost from the genome assembly (either completely or partially). This in turn, poses a limitation for the reconstruction of metabolic models using these assemblies, since most published approaches use sequence searches to predict the presence/absence of genes and their associated enzymatic reactions. Therefore, if we are to use public genome data for high-throughput metabolic modelling studies, it is essential to evaluate the expected model accuracies and understand the minimum input genome quality requirements.”

The biological accuracy of the curated ‘complete’ models has been described previously, and this is now noted in the text at lines 320-324:

“Here we performed a systematic analysis leveraging our published curated KpSC models (n=37, (14)), which were generated using completed genome sequences and were therefore considered to represent ‘complete’ models for which the underlying genome sequence contains all genes that are truly present in the genome (note the biological accuracy of these models was reported previously (14) and is not the subject of the current study).”

Throughput the manuscript we not only compare models in terms of the numbers of genes and reactions, but through comparison of binary growth predictions. Specifically, in the Performance Comparison section (Bactabolize vs other approaches) we use comparison of predicted to experimental phenotypes for strain KPPR1 (see response to rev 1 point 4 for details). In the QC Framework section we compare the predictions derived from draft models generated from draft genome assemblies to those derived from the matched ‘complete’ models, and report the concordance as a measure of impact of input assembly quality (lines 309-394). In the final results section (Predictive accuracy of draft models), we generate 10 additional models and compare the growth predictions to matched experimental data (lines 414-433). We view these phenotype prediction comparisons as the ultimate measure of ‘completeness’ with which to assess our models, because these data have direct biological meaning.

1. The authors have not provided evidence or discussion on the accuracy of any metabolic fluxes, which are considered to be crucial for reconstructing metabolic models. Additionally, the authors have not mentioned the importance of non-growth associated maintenance and the criticality of biomass composition analysis, both of which significantly determine the fluxes in the system.

We acknowledge the importance of flux calculations and accurate biomass compositions when using genome-scale models to quantitatively predict growth rates. However, at this stage, the reconstructions developed using Bactabolize are intended for binary predictions and comparisons of growth capabilities on various substrates. The accuracies we report are based on measures of network completion (presence/absence of relevant reactions leading to growth or no-growth phenotypes) rather than specific growth rates. Thus, the models generated by Bactabolize can be used to explore diversity at the strain level in terms of growth capabilities and can serve as a scaffold for building detailed (customized biomass), strain-specific models. Measuring biomass composition and metabolic flux analysis require significant experimental comparisons that are outside the scope of the current study but could be performed for target strains based on reconstructions developed using Bactabolize.

1. It would be interesting to compare the accuracy of the models generated using Bactabolize with those manually curated.

We did exactly this. We compared the manually curated model iKp1289 as part of our benchmarking. Lines 194 – 199:

“We compared the output and performance of Bactabolize to the two previously published tools that can support high-throughput analyses i.e. CarveMe (30) and gapseq (31). To aid interpretation in the context of community standard approaches, we also include a comparison to the popular web-based reconstruction tool, KBase (ModelSEED), and a manually curated metabolic reconstruction of K. pneumoniae strain KPPR1 (also known as VK055 and ATCC 43816, metabolic model named iKp1289) (15).”

Unfortunately, as far as we aware there are currently no other published manually curated models for strains with matched phenotype data that are also not included as part of our pan-reference model (the latter is a key point to ensure a fair comparison of models generated using our pan-reference vs those generated with a universal reference).

1. The authors have not provided evidence or discussion on the accuracy of any metabolic fluxes, which are considered to be crucial for reconstructing metabolic models.

See response to rev 3, point 2.

1. The justification regarding the completeness of the models requires further discussion.

See response to rev 3, point 1.

1. A detailed discussion on the importance of manually curated models would significantly enhance the quality of the manuscript.

This has been added at lines 458-474:

“Traditionally, genome-scale metabolic reconstruction approaches have relied upon significant manual curation efforts. While there will always remain a need for high quality curated models, such resource intensive approaches preclude their application at scale, and have therefore limited analyses to small numbers of individual strains (15, 16). However, automated reconstruction approaches can support the generation and comparison of multiple strain-specific draft models from which meaningful biological insights can be derived (61). Additionally, the quality of curated models is likely to vary depending on their age, level and type of curation, as well as the approach used for preliminary drafting. Indeed it is possible for automated approaches to outperform manually curated models; a draft model for K. pneumoniae KPPR1 generated using Bactabolize with the KpSC pan-v1 reference model outperformed the manually curated iKp1289 model representing the same strain (15). iKp1289 was published in 2017 (6 years prior to this study) and was initially drafted via the KBase pipeline (33), which uses RAST to annotate the sequences with Enzyme Commission numbers. It has been demonstrated several times that the Enzyme Commission scheme has systematic errors (62, 63), leading to a loss in accuracy when compared to the ortholog identification methods used by automated approaches. Consistent with this assertion, our draft KPPR1 model constructed with KBase (without manual curation) was an outlier in terms of the very low number of genes, reactions and metabolites that were included.”